# Extended-Spectrum Beta-Lactamases Producing *Enterobacteriaceae* in the USA Dairy Cattle Farms and Implications for Public Health

**DOI:** 10.3390/antibiotics11101313

**Published:** 2022-09-27

**Authors:** Benti Deresa Gelalcha, Oudessa Kerro Dego

**Affiliations:** Department of Animal Science, University of Tennessee, Knoxville, TN 37996, USA

**Keywords:** extended-spectrum beta-lactamase, *Enterobacteriaceae*, beta-lactam antibiotic, dairy cattle, public health, antimicrobial resistance, antibiotic resistant bacteria, antibiotic resistance gene

## Abstract

Antimicrobial resistance (AMR) is one of the top global health threats of the 21th century. Recent studies are increasingly reporting the rise in extended-spectrum beta-lactamases producing *Enterobacteriaceae* (ESBLs-Ent) in dairy cattle and humans in the USA. The causes of the increased prevalence of ESBLs-Ent infections in humans and commensal ESBLs-Ent in dairy cattle farms are mostly unknown. However, the extensive use of beta-lactam antibiotics, especially third-generation cephalosporins (3GCs) in dairy farms and human health, can be implicated as a major driver for the rise in ESBLs-Ent. The rise in ESBLs-Ent, particularly ESBLs-*Escherichia coli* and ESBLs-*Klebsiella* species in the USA dairy cattle is not only an animal health issue but also a serious public health concern. The ESBLs-*E. coli* and *-Klebsiella* spp. can be transmitted to humans through direct contact with carrier animals or indirectly through the food chain or via the environment. The USA Centers for Disease Control and Prevention reports also showed continuous increase in community-associated human infections caused by ESBLs-Ent. Some studies attributed the elevated prevalence of ESBLs-Ent infections in humans to the frequent use of 3GCs in dairy farms. However, the status of ESBLs-Ent in dairy cattle and their contribution to human infections caused by ESBLs-producing enteric bacteria in the USA is the subject of further study. The aims of this review are to give in-depth insights into the status of ESBL-Ent in the USA dairy farms and its implication for public health and to highlight some critical research gaps that need to be addressed.

## 1. Introduction

Antimicrobial resistance (AMR) is one of the most critical global health challenges [1,2]. Globally, about 700,000 deaths were attributed to diseases caused by antibiotic resistant organisms. The major concern is that if proper intervention measures are not implemented, this figure is predicted to rise to 10 million deaths annually in 2050 [2]. Every year, in the USA alone, 2.6 million people suffer from infections caused by antibiotic resistant bacteria (ARB), and about 17% of them die [3]. The drivers of AMR emergence are complex and multifactorial. But the widespread use and misuse of antibiotics in livestock and human medicine are recognized as the leading driver of AMR [4]. 

Antimicrobials are widely used in food-producing animals throughout the globe. Intensive food animal production systems such as dairy, beef, poultry, and swine productions frequently use medically important antimicrobials (MIAs) for therapeutic, prophylactic, and metaphylactic purposes [5,6]. For instance, antibiotics, regarded as the highest priority and critically important (e.g., third-generation cephalosporins-3GCs) for treating human infections that are refractory to other antibiotics, are widely used in dairy farms for the prevention and treatment of various diseases in dairy cattle [6,7,8]. Cattle carry many bacteria in the group of *Enterobacteriaceae* in their gastrointestinal tract, which are frequently exposed to these critically important classes of antibiotics (CIAs) and MIAs [9]. Critically important antibiotics (CIAs) are antimicrobial drugs used to treat enteric pathogens that cause foodborne disease and a last -resort therapy or one of few alternatives to treat serious human disease where first-line antibiotics have not worked. Medically important antibiotics (MIAs) are antibiotics that are important for treating human diseases including critically important, highly important and important antibiotics [10].

The continuous exposure of *Enterobacteriaceae* to CIAs such as 3GCs can lead to the selection and spread of ARB and their antimicrobial resistance genes (ARGs). The ARB and ARGs could spread to humans through direct contact or indirect routes [11,12,13,14]. Extended-spectrum beta-lactamases (ESBLs) encoding genes mediate resistance to third- and sometimes to fourth-generation cephalosporins, the “highest priority and CIAs’’ [7,15]. So the rise in the incidence of ESBLs-Ent such as *E. coli* and *Klebsiella* spp. in dairy farms is of significant public health concern since these antibiotics are the highest priority and critically important ones for the treatment of human infections caused by Gram-negative bacterial pathogens [16].

Ceftiofur, a 3GC, is one of the top three most frequently used antibiotics to treat and prevent mastitis and other diseases of dairy cattle [5,17]. Recent studies indicated that resistance to beta-lactam antibiotics, specifically resistance to 3GCs, is rising among commensal *Enterobacteriaceae* isolates from the USA dairy cattle [18]. Resistance to 3GCs is mainly mediated by the production of ESBLs, a group of enzymes that break down a beta-lactam ring of the extended-spectrum cephalosporins such as 3GCs [19]. Among *Enterobacteriaceae*, *E.*
*coli* and *Klebsiella* spp. are among the most frequently identified bacteria carrying ESBL-encoding genes, such as *bla*_CTX-M_, *bla*_SHV_, and *bla*_TEM_ [20,21,22,23,24]. Ceftriaxone and cefotaxime are similar 3GCs antibiotics used to treat severe infections caused by pathogenic strains of *Enterobacteriaceae* in humans [25,26]. The use of the same generation of cephalosporins with the same chemical structure, active ingredients, and spectrum of activity in dairy cattle farms and human health settings may lead to cross-resistance that can be transferred to humans or vice versa via direct and indirect routes [27,28,29]. The aims of this review are to give a detailed account of the current status of ESBL-Ent in the USA dairy cattle farms and its implication for human health and highlight research gaps that need to be addressed.

## 2. Mechanisms of Resistance to Beta-Lactam Antibiotics

Beta-lactam antibiotics including 3GCs such as ceftiofur, ceftriaxone, and cefotaxime, are widely used against Gram-negative pathogens [30]. These antibiotics act by covalently binding to penicillin-binding proteins (PBP), an enzyme that catalyzes the polymerization and transpeptidation of peptidoglycan [31]. The binding of antibiotics to PBP will lead to their inactivation and thereby inhibition of cell wall synthesis and death of susceptible bacteria [31,32].

*Enterobacteriaceae* resistance to 3GCs has become an alarming and growing public health challenge [3,21]. Bacteria employ three resistance mechanisms against beta-lactam antibiotics. These include (1) mutations that change the structure of penicillin-binding proteins, (2) change in cell permeability (disruptions of porin proteins in the outer membrane or increase in efflux pumps) and (3) production of beta-lactamase enzymes, which hydrolyzes the beta-lactam ring in beta-lactam antibiotics [33,34,35].

In *Enterobacteriaceae*, resistance to 3GCs is primarily mediated by the production of beta-lactamases [36]. In addition to ESBLs, resistance to extended-spectrum beta-lactam antibiotics could be mediated by carbapenemase (encoded by *bla*_KPC_, *bla*_NDM_, *bla*_OXA-48_, etc.), plasmidic AmpC (pAmpC; commonly encoded by the *bla*_CMY_ genes), and mutations in AmpC promoter regions in the chromosome [20,37]. However, this review focuses on ESBLs, as ESBL production is one of the most important and common resistance mechanisms employed by *Enterobacteriaceae* against extended-spectrum beta-lactam antibiotics [21,38]. The definition of ESBLs is ambiguous, and in this review, we adopt a more comprehensive ESBLs definition as beta-lactamases that hydrolyze or confer resistance to penicillins, cephalosporins (First- to third-generations), monobactams (e. g., aztreonam) but not the cephamycins (e.g., cefotetan and cefoxitin) or carbapenems (e.g., meropenem and imipenem) and are inhibited by beta-lactamase inhibitors (e.g., clavulanate) [39,40].

Although ESBLs share common biochemical properties, they all break down extended-spectrum beta-lactam antibiotics and are inhibited by clavulanate; the genes encoding these enzymes are diverse [41,42]. The most frequent variants of ESBLs include the CTX-M (cefotaxime-hydrolyzing beta-lactamase), SHV (sulfhydryl reagent variable), and TEM enzymes. The TEM was determined initially in a single strain of *E. coli* isolated from blood of a patient [43,44]. The parent type of SHV (SHV-1) and TEM (TEM-1, 2) are narrow-spectrum beta-lactamases that give rise to their respective ESBL variants through mutations [45]. Amino acid substitutions or mutations in the genes encoding these enzymes give rise to expanded substrate specificity or enhanced hydrolytic activity [41,46]. As a result, the number of identified variants of the TEM and SHV families is continuously rising, most of which have emerged and are also currently emerging via stepwise mutations [41].

TEM-1, the first TEM type beta-lactamase, was first reported almost six decades ago (in 1965) in Greece from a patient named Temoneira, from which it was designated as TEM [40,43]. This enzyme variant resists narrow-spectrum beta-lactam antibiotics (penicillin and the first-generation cephalosporins before it expanded its spectrum via mutations [40].

The SHV enzymes were first reported nearly five decades ago (in the 1970s), and the first reported variant (SHV-1) exhibited activity against the penicillins and first-generation cephalosporins [45,47]. In the 1980s, the SHV- and TEM-ESBL variants mutants of the parent enzymes, were the prominent cause of resistance to 3GCs among *Enterobacteriaceae* [48]. As of 5 July 2022, 229 *bla*_SHV_ and 246 *bla*_TEM_ variants have been reported [49]. 

As opposed to the TEM- and SHV-ESBLs variants, CTX-M (cefotaximase) type enzymes did not evolve from mutations of existing enzymes; the gene encoding this enzyme was acquired from *Kluyvera* spp. through horizontal gene transfer [50]. After mobilization of this gene from the chromosome into a plasmid, mutations lead to further diversification and provide the opportunity for expansion of hydrolytic activity to other extended-spectrum cephalosporins such as ceftazidime [51]. In animals, the CTX-M enzyme was first identified in 1988 from *E. coli* isolated from dogs’ feces in Japan [52]. Since the 2000s, CTX-M-ESBL variant has become the most prevalent and widespread cause of resistance to extended-spectrum cephalosporins among the *Enterobacteriaceae* across the globe, both in humans and food animals such as dairy cattle [21,24,38,53,54,55,56,57]. Based on the order of identification of the group’s founder and amino acid sequence identity, the CTX-M family of ESBLs are phylogenetically categorized into five distinct groups designed as 1, 2, 8, 9, and 25 [53,56]. Each of the five groups differs by at least 10% amino acid sequence identity [58]. There are several minor variants within each group, and currently, at least two groups of CTX-M (group 1 and 9) are described in USA dairy farms [24]. As of July 5, 2022, 252 *bla*_CTX-M_ ESBL variants have been described [59] 

## 3. Use of Beta-Lactam Antibiotics in the USA Dairy Cattle Farms

Beta-lactam antibiotics are the most frequently used class of antibiotics characterized by the beta-lactam ring, a similar biochemical structure across the class [32]. These include penicillins, cephalosporins, monobactams, cephamycins, carbapenems, and beta-lactamase inhibitors [34]. Beta-lactam antibiotics are the most frequently prescribed class of antibiotics in human health [32] and on dairy farms [5,17]. In the USA, cephalosporins and penicillins are the most commonly used beta-lactam antibiotics to treat or prevent mastitis and other common diseases of dairy cattle [17,60]. According to the USA Food and Drug Administration (FDA) 2019 report, from a total of 29,830 kg of cephalosporins sold and approved for use in food-producing animals, the vast majority (81%) were distributed to cattle production [6]. Similarly, Nora et al. [61] also reported that the largest amount of cephalosporins (10.5 g per cow year) and penicillins (4.49 g per cow year) are used in dairy cattle farms compared to other classes of antibiotics whose use is less than 1 g per cow year.

Ceftiofur and cephapirin are the only cephalosporins licensed for use in the USA food animals, including dairy cows [62,63]. Cephapirin is only approved as an intramammary infusion for the treatment of mastitis caused by *Streptococcus* and *Staphylococcus* species [64]. Cephapirin is the active ingredient in cephapirin sodium (Brand name: Today) is used for the treatment of mastitis in lactating cows and cephapirin benzathine (Brand name: Tomorrow) is used for the treatment of mastitis in dry cows. Ceftiofur is approved in two forms, injectable and intramammary infusion. In the USA, three parenteral and two intramammary formulations of ceftiofur are approved for use in dairy cattle. The two intramammary formulations of ceftiofur are (1) Ceftiofur hydrochloride suspension for treatment of lactating cows (Brand name: SPECTRAMAST LC) and (2) Ceftiofur hydrochloride suspension for treatment of dry cows (Brand name: SPECTRAMAST DC). The parenteral formulations include (1) ceftiofur sodium (Brand name: Naxcel), (2) ceftiofur hydrochloride (Brand name: Excenel), and (3) ceftiofur crystalline-free acid (Brand name: Excede). These all formulations are approved for treating bovine respiratory disease (BRD) and footrot but with different treatment regimens. In addition, ceftiofur hydrochloride (Brand name: EXCENEL RTU- Excenel ready to use) and ceftiofur crystalline-free acid are used for the treatment of acute metritis at different dosage regimens. Ceftiofur hydrochloride suspensions for lactating (Brand name: SPECTRAMAST LC) and dry (Brand name: SPECTRAMAST DC) cows are used for treating clinical mastitis caused by non-aureus staphylococci also known as coagulase-negative *Staphylococcus* species (CNS), *Streptococcus dysgalactiae (S. dysgalactiae)*, and *E. coli* during lactation and subclinical mastitis caused by these bacteria at the time of drying off [64,65,66,67], respectively.

The most common infectious diseases of dairy cattle treated with antibiotics include mastitis, lameness, and respiratory and digestive diseases [5,68]. In dairy cattle, ceftiofur is indicated to treat mastitis, bovine interdigital necrobacillosis, bovine respiratory disease, acute postpartum metritis, and mastitis caused by coliform bacteria. Ceftiofur is the most widely used 3GCs for the prevention and treatment of mastitis in dairy cattle [45,46,47,65]. The recent USA National Animal Health Monitoring System (NAHMS) survey report showed that the highest proportion (27.6%) of pre-weaned heifers and about 7.2% of weaned heifer calves were treated for diarrhea with ceftiofur as a primary antibiotic. Similarly, 10.3% of pre-weaned heifers and 13.4% of weaned heifers were given ceftiofur as a primary antibiotic for treating BRD [17]. A relatively recent survey on several dairy herds in the USA also reported frequent use of ceftiofur to prevent and treat respiratory and digestive diseases in dairy calves [5,69].

Almost all studies found that mastitis is the primary reason for the use of antibiotics in dairy cattle [5,70,71,72,73]. According to NAHMS, ceftiofur is a primary antibiotic used to treat 50.5% of mastitis cases, 45.6% of reproductive disorders, 58.7% of lameness, 77.6% of respiratory infections, and 57.4% of digestive tract diseases [17]. The broad spectrum of activity and the shorter milk withdrawal period make ceftiofur the most popular and ideal antimicrobial drug for dairy cows [74]. The cephalosporins use in USA food-producing animals (cattle, swine, turkey, and chickens) has shown a general increasing trend in the last decade (Figure 1A).

The cephalosporins sold for cattle use were declined from 2016 to 2017, followed by an increase from 2017 to 2018 and a declining trend from 2018 to 2019 (Figure 1B).

Accurate data on cephalosporins used in USA dairy cattle are not available. The USA Food and Drug Administration Center for Veterinary Medicine report showed that the total quantity of cephalosporins sold and distributed in the USA for cattle use (Figure 1B). However, the actual amount of cephalosporins used in dairy and beef cattle farms is not explicitly described and thus it is unknown [6]. The absence of actual quantity of cephalosporins administered or given to dairy cattle is one of the biggest challenges in assessing the impact of its use on the emergence of resistance to this class of antibiotics.

A recent study on *Klebsiella* isolates from cases of mastitis indicated that the prevalence of ceftiofur-resistant *Klebsiella* spp. was low from 2008 to 2016 and abruptly increased between 2016 and 2017 and then decreased in 2019 (Figure 1B). The increase in prevalence could be related to the rise in the use of ceftiofur (Figure 1B) for prevention and treatment of mastitis and other diseases of dairy cattle during the specific year, which might have increased in response to the increased selection pressure on *Klebsiella* spp. [76].

Both experimental and observational studies indicated that antibiotic use and thus, the resultant selection pressure is an important factor driving the emergence and persistence of ARB and their resistance genes [77,78,79,80,81,82,83,84]. Mastitis is the most frequent disease of dairy cattle; and ceftiofur is the most commonly used antimicrobial drug used to manage it [5,68]. This implies mastitis is an important indirect driver of the emergence of ESBL-producing bacteria in the USA dairy farms [6,9,20]. However, despite increased usage of ceftiofur in dairy farms, several studies examining the AMR status of mastitis pathogens in dairy cattle did not find a rise in the prevalence of resistance to ceftiofur [85,86,87,88,89]. Nevertheless, it should be noted that AMR among commensals and foodborne pathogens from dairy cows and dairy manure is increasing [90]. Our published and ongoing studies also showed an increased prevalence of ESBLs-Ent (e.g., *E. coli*, *Klebsiella* spp.) from dairy manure and bulk tank milk [18].

Thus, heavy reliance on ceftiofur for the prevention and treatment of mastitis should be reassessed, and prudent and antimicrobial stewardship-focused utilization of ceftiofur is important to reduce the development of resistance against this CIA. The use of other alternative dairy cattle disease control measures such as good hygienic and biosafety practices, use of teat sealants at drying off, vaccines, and full implementation of mastitis control plan with good plan of nutrition and management should be considered to reduce the incidence of mastitis and other diseases of dairy cattle [91,92,93,94]. A study showed that blanket application of teat sealants to all cows at drying off had a protective effect against the emergence of ESBLs at the herd level [92]. Thus, the widespread use of this method may help reduce cephalosporin use and thereby emergence of ESBLs-Ent. 

## 4. Molecular Epidemiology of Extended-Spectrum Beta-Lactamases Producing *Enterobacteriaceae* in the USA Dairy Farms

The epidemiology of ESBLs-Ent is complex and rapidly changing [4,95]. The epidemiology of ESBL genes is influenced by horizontal transfer of resistance genes, presence of additional resistance gene/s, rapid mutation of existing resistance genes to generate new variants, expansion of bacterial host range carrying the gene, and sometimes convergence of ESBL genes and virulence genes in the host bacteria [38,96].

In *Enterobacteriaceae,* ESBL genes are primarily associated with mobile genetic elements (MGEs) such as plasmids, insertion sequences (IS), transposons, integron cassettes, and prophages-mediated intracellular and intercellular movement [97]. The MGEs play a vital role in the spread of ESBL genes; for example, IS, transposons, and integron cassettes may mediate the movement of ESBL genes within the same bacterial genome, whereas plasmids can transfer genes between different bacteria cells [97,98].

IS*Ecp1*, an insertion element situated upstream of the CTX-M-encoding gene, causes the movement of this ESBL gene from the chromosome of *Kluyvera* spp. onto plasmid [99,100]. Then, the plasmid can spread to other populations of bacteria such as *E. coli* and *Klebsiella* spp. [95,98]. Some ESBLs-harboring plasmids have a broader host range, whereas others have a limited or restricted host range. Broad-host range plasmids are characterized by their ability to replicate and easily transfer between different species of bacteria. Thus, enhancing interspecies transmission and spread of ESBL genes and other ARGs [98,101]. These groups of plasmids include those belonging to the family of IncA/C, IncI, IncN, IncHI2, IncL/M, IncK, and IncN [95,98].

In contrast, narrow-host range plasmids tend to be restricted to certain species or strains within a given species. Thus, their role is also limited to intraspecies dissemination of ESBL genes [101,102]. Narrow-host range plasmids play a crucial role in disseminating ESBL genes, particularly the CTX-M-variants [101]. For instance, IncF plasmids, also known as “epidemic resistance plasmids, “have a strong tendency to acquire and disseminate ESBL genes and other ARGs among members of *Enterobacteriaceae* [103].

Some studies reported the worldwide distribution of *bla*_CTX-M-15_ gene variants of ESBL is due to its frequent association with IncF plasmids [24,38,95,104]. Molecular epidemiological studies have shown that these plasmids frequently harbor additional genes involved in fitness, virulence, and other ARGs (e.g., *qnr*, *qepA*, *tetA*, *floR*, *sul2*, and *cmlA*) that may help the host bacteria to survive and thrive in human and animal hosts [18,96,105,106,107]. *E. coli* ST131 is the classic example of the most successful ESBLs-producing pathogenic strains associated with IncF plasmids. The high-risk clone of ESBLs-*E. coli* has been reported in humans and animals, including dairy cattle in the USA, Canada, Spain, Korea, and India [24,101,103,104,108]. It has been suggested that the successful spread of this virulent strain of *E. coli* across the globe is related to its association with the IncF plasmids, particularly the FIA and FII replicon types [103,108,109].

Molecular studies have consistently shown that ESBL genes in *Enterobacteriaceae* isolates are physically associated with MGEs that carry other resistance genes mediating resistance to an unrelated class of antibiotics [110,111]. Co-resistance to multiple classes of antibiotics such as fluoroquinolones, tetracyclines, chloramphenicol, aminoglycosides, sulfonamides, and /or trimethoprim has been frequently reported in *E. coli* and *Salmonella* species isolated from dairy cattle and other food-producing animals (e.g., in Pigs and poultry) in the USA and elsewhere (e.g., India, China, and Egypt [18,24,111,112,113,114,115].

Among the ESBL families, CTX-M-type is rapidly evolving and spreading in the USA and worldwide [15,21]. Currently, CTX-M- genes appear to be the dominant type of ESBL genes in *Enterobacteriaceae*. The cause of the dominance of the CTX-M variant of ESBL over the TEM and SHV variants is not clearly understood. Some authors speculated that the success of *bla*_CTX-M_ might be related to the less fitness cost of expressing CTX-M enzymes in the host bacteria, more effective mobilization of the gene by MGEs, and selection pressure from antibiotics [24,58].

This ESBL-variant is mainly associated with multidrug resistance (MDR) encoding plasmids [116]. MDR CTX-M-producing *E. coli* in USA dairy cattle and dairy cows’ manure have been shown to be frequently related to transferable resistance genes to other critically important and highest priority classes of antibiotics. For instance, plasmid-mediated fluoroquinolone resistance genes (e.g., *qnr*) and macrolides resistance gene (e.g., *mph*(A)) have been reported together with CTX-M genes in *E. coli* isolated from the USA adult dairy cattle, dairy calves, and dairy cows’ manure [8,18,115]. As a result, detection of ESBL genes in *Enterobacteriaceae* is considered a hallmark of MDR. This MDR phenotype poses a serious threat to public health if the commensal *E. coli* passes the resistance gene cassettes to other enteric human pathogens such as *Salmonella* spp. or pathogenic strain of *E. coli* or *Klebsiella* spp. [115].

## 5. Emergence and Status of Extended-Spectrum Beta-Lactamases Producing *Enterobacteriaceae* in the USA Dairy Farms

In the USA, the occurrence of ESBLs-Ent has long been reported in humans and animals, including dairy cattle. In humans, the TEM family ESBLs was first described in 1987 [117] and the SHV-type ESBLs in late 1980 from Boston, Massachusetts [118]. The CTX-M type ESBL was first described in 2003 from *E. coli* isolates obtained during hospital surveillance in five States (Virginia, Idaho, Ohio, Washington, and Texas) [119].

In the USA dairy cattle, the occurrence of ESBLs-Ent was first reported from Ohio more than two decades after it was reported in humans [27]. Since then, studies have shown that the number of bacterial isolates resistant to 3GCs is rapidly on the rise in dairy cattle. Widespread use of ceftiofur has been linked to increased isolation of 3GCs-resistant fecal bacteria from dairy cattle. In the USA, ESBLs-*E. coli* and -*K. pneumoniae* have been isolated from milk of cows with mastitis, bulk tank milk [18,76], rectal fecal samples, lagoon, and soil amended by cow manure [18,24]. This highlights that ESBLs-*E. coli* and -*K. pneumonia* may enter the food supply through contaminated milk and beef products of dairy cows’ origin.

In the USA, few studies have investigated the status of ESBLs-Ent in dairy cattle, dairy cattle-derived food products, and their environment [18,24,27,27,57,120,121,122]. A recent global review of the status of ESBLs-Ent in cattle clearly showed the availability of very limited information on the prevalence of ESBLs-Ent, including *E. coli* and *Klebsiella* spp. in USA dairy farms [123]. Furthermore, comparing findings from these studies is difficult because these studies vary widely in terms of design, sampling methods, sample types, and ESBLs detection methods (Table 1). As a result, data from these studies may not be generalizable and, thus, not possible to determine the status of ESBLs-Ent in the dairy cattle population. This impedes proper estimation of the public health risk arising from ESBLs-Ent in the dairy farm, affecting the design and implantation of appropriate intervention measures.

Previous studies often focused on phenotypic detection of resistance to 3GCs in *Enterobacteriaceae*, mainly in *E. coli* and *Salmonella* spp. in dairy farms. Only recently, a few studies attempted to identify the genetic determinants of the observed phenotypes. Most earlier reports indicated *bla_CMY-2_* genes as dominant genes mediating resistance to extended-spectrum cephalosporins in *Enterobacteriaceae* in the USA dairy farms while ESBL genes were reported relatively recently [133]. Recent reports suggested that *bla*_CTX-M_ genes are the most prevalent ESBL genes responsible for resistance to extended-spectrum cephalosporins among *Enterobacteriaceae* in the USA dairy farms (Table 2).

Among *bla*_CTX-M_ variants, *bla*_CTX-M-15_, which belongs to CTX-M group 1, was the most prevalent allele reported in USA dairy farms. The gene is frequently associated with the IncI1 plasmid replicon type, an easily transmissible narrow-host-range plasmid that might have contributed to its widespread. In addition to ncI1, ESBL genes reported from USA dairy farms were associated with other plasmid replicon types such as IncFIB, IncF, IncFIA/FIB, IncN, IncFIA, IncFIA/11, and IncB/O, IncA/C, IncN.

In addition to studies from various USA Universities and other research institutes, the USA National Antimicrobial Resistance Monitoring System (NARMS) survey also reported different kinds of beta-lactamase encoding genes in *E. coli* and *Salmonella* species isolated from the USA dairy operations [134]. The NARMS report (Table 3) showed that the majority of beta-lactamase genes detected in *E. coli* was *bla*_TEM-1_ (69%). In contrast, the most frequent and widespread beta-lactam antibiotics resistance gene detected in *Salmonella* isolates of dairy origin was *bla*_CMY-2_ (72.6%).

In addition to *E. coli, Klebsiella*, and *Salmonella* spp., other ESBLs-Ent members such as *Enterobacter* spp. could be important both as a mastitis-causing pathogen and as a possible vector of ESBL genes. To the authors’ knowledge, there is no at least published report of ESBL-producing *Enterobacter* spp. from USA dairy farms. However, detailed studies are required to determine the contribution of Enterobacter species to the ESBLs-Ent.

## 6. Public Health Implications of the Rise in Extended-Spectrum Beta-Lactamases in Dairy Cattle Farms

The association between levels of ESBLs-*Ent* in cattle and its occurrence in humans is complex and could be influenced by bacterial strains, MGEs, and the frequency of direct and indirect interaction among cattle, humans, and environments [23,135]. Previous studies have shown that farm workers are more likely to be colonized by multidrug-resistant *E.coli* than people without direct contact with animals [136]. Thus, direct exposure to dairy cattle and farms may provide an important mechanism for spreading ESBLs-Ent from dairy farms to the community. In dairy cattle, *Klebsiella* spp. *(K. pneumoniae, K. oxytoca)* and *E. coli* are the most common coliform bacteria that cause bovine mastitis [76,137,138]. However, the strains that cause bovine mastitis and those that cause human infections may not be the same. However, some strains of ESBLs-*E. coli* and -*K. penumoniae of* dairy origin can pass to humans or transfer their resistance genes to human pathogenic strains. In humans, pathogenic strains of ESBLs-*E. coli* and -*K. penumoniae* are associated with hospital-acquired and community-acquired urinary tract and bloodstream infections [3,139]. As previously mentioned, non-pathogenic strains of these bacteria may pass the ESBL genes to human pathogenic strains causing cross-resistance to other 3GCs such as ceftriaxone and cefotaxime, which are considered critically important for the treatment of human infections [7].

Currently, ESBLs-Ent infections are on the rise in humans [140,141]. The USA CDC reported continuous increases in community-associated human infections caused by ESBLs-Ent. This report showed a 9% average annual increases in the number of hospitalized patients from ESBLs pathogens in five consecutive years (Figure 2), an estimated 197,400 cases of ESBLs-Ent among hospitalized patients, and 9100 estimated deaths and an estimated USD 1.2 billion health care costs in the USA in 2017 alone [3].

The sources of community-acquired ESBLs-Ent infections are unknown and can be from multiple sources; a system-based study is required to understand the contribution of different sources. Some researchers believe that extensive use of ceftiofur in production animals, such as in dairy farms, is a risk factor for rising ESBLs-Ent infections in humans in the USA [116,142]. This argument is acceptable to a certain extent, as previous studies have found that the therapeutic use of antibiotics in animals could increase the prevalence of antibiotic resistant *Enterobacteriaceae* such as *E. coli* in animals and its risk of transmission to humans [143,144]. However, only a few epidemiological studies have demonstrated strong evidence of transmission of 3GCs-resistant *Enterobacteriaceae* from cattle to humans in the USA [74,121,145,146]. Nevertheless, it should be noted that the lack of strong evidence of ESBLs-Ent transmission occurring between dairy cattle and humans may not necessarily suggest the absence of transmission. It might be related to a lack of utilization of high-resolution molecular techniques such as WGS, lack of sensitivity of sampling (inability to detect ESBLs-Ent from diverse populations in a sample), transient colonization of ESBLs-Ent, which may escape detection at the time of sampling, and possible recombination events or increased mutation rate [147,148]. Robust data showing transmission of ESBLs-Ent, or ESBLs genes from cattle to humans is of great importance to develop appropriate intervention measures and policies on beta-lactam antibiotics use and stewardship and infection control in dairy cattle and other farm animals [149]. The use of WGS technologies along with an appropriate study design (that shows temporal and spatial connection) are critical tools to generate valid inferences on the extent of ESBLs-Ent and ESBL genes transmission at dairy cattle-human interfaces as well as the spread among dairy farms, humans and the environments [23,149].

Two mechanisms of spread of ESBL genes are expected among *Enterobacteriaceae* in dairy farms and humans. These are (1) clonal and (2) horizontal spread of ESBL genes [135]. The ESBLs-Ent and ESBL genes can transmit from cattle to humans through direct or indirect routes (Figure 3). Direct transmission involves close contact between animals and humans (hand to mouth). Indirect transmission can occur via food chain such as food of animal origin (consumption of unpasteurized raw milk, undercooked meat, and unpasteurized fresh fruits and vegetables contaminated with ESBLs-Ent) or via contaminated environmental sources such as soil, crops and surface water (Figure 3) [149].

The gastrointestinal tract of dairy cattle contains many enteric bacteria, such as *E. coli*, *Klebsiella*, *Salmonella* species and others. Ceftiofur use in dairy cattle selects for ceftiofur-resistant *Enterobacteriaceae*, which might contaminate food of dairy cattle origin (milk and ground Beef) and eventually pass to humans. ESBLs-Ent can also enter the food supply as contaminants of fresh leafy vegetables from manure or wastewater used to fertilize fruits and vegetables. Dairy farm workers and individuals working in agricultural farmlands have a higher risk of exposure to dairy cattle harboring ESBLs-Ent or their manure containing ESBLs-Ent. Thus, they may become carriers and transmit the resistant bacteria to their close contacts (household members) and the broader community (Figure 3). ESBLs-Ent may be maintained in the dairy farm through oral-fecal transmission via contaminated feed and water [150].

Ingestion of ESBLs-bacteria such as *E. coli and Klebsiella* spp. via direct or indirect routes may result in the colonization of the human gastrointestinal tract. Upon colonization, based on its pathogenic potential, the growth of ESBL-producing bacteria could lead to an infection or persistence as a commensal. Both conditions will allow ESBL-producing bacteria to spread to other humans clonally [151] or horizontally transfer ESBL genes to other bacteria via MGEs [152].

### 6.1. Unpasteurized Milk and Undercooked Beef as Possible Sources of Extended-Spectrum Beta-Lactamases Producing Enterobacteriaceae

It is well-recognized that unpasteurized milk is a source and vehicle for transmitting several bacteria of animal origin [153,154,155]. More recently, a study estimated that raw milk is consumed by only 3.2% of the population, and cheese is consumed by only 1.6% of the population, but responsible for 96% of diseases caused by contaminated dairy products in the USA [156]. The same group of researchers estimated that the odds of unpasteurized milk and milk products causing illness is 840 times that of pasteurized products. Similarly, a study conducted over 14 years indicated that three-fourths of outbreaks linked to dairy products occur in states that allow the sale of raw milk [157].

Commensal bacteria that acquire ESBL genes in dairy farms may spill over to humans indirectly through the consumption of unpasteurized milk and milk products [158]. Consumption of these products may favor mixing these bacteria with the human enteric microbiota or other more pathogenic strains of bacteria [159]. The mixing of ESBLs-producing bacteria with pathogenic strains enhances the chance of ESBL genes being shared through horizontal gene transfer, creating a threat to human health [18,145,160]. In our recent study, we detected cefotaxime resistant *E. coli* in bulk tank milk [18]. All cefotaxime resistant *E. coli* isolates identified in this study carried *bla*_CTX-M_ type ESBL gene together with other multiple resistance genes conferring resistance to several classes of antibiotics [18]. Similarly, a previous study reported that 3GCs resistant *E. coli* and *Klebsiella* species isolates from bulk tank milk had MDR phenotypes [159].

In addition to the risk of transferring ESBL genes to other clinically important strains, some members of *Enterobacteriaceae*, such as *E. coli* and *Salmonella* species are reported to be among the most important milkborne human pathogens in the USA [161]. For instance, a study reported an outbreak of *E. coli* O157:H7 infections caused by consumption of unpasteurized milk from a specific dairy farm in Portland, Oregon. The study reported homologous strains (pulsotypes) of *E. coli* O157:H7 isolates from human cases and the dairy herd where milk had originated [162]. *E. coli* causes a wide range of severe infections in humans [163] and is also among the frequent causes of environmental mastitis in dairy cattle [164], along with colibacillosis in calves [165]. 

Some reports indicated that more than 30 states in the USA allow the legal sale of raw milk [166]. Previous data also showed that most foodborne outbreaks are reported from states allowing the sale of raw milk [161]. The rise in the occurrence of MDR ESBLs-producing bacteria such as *E. coli*, along with the increase in legalization and consumption of raw milk, will create a high-risk synergy that jeopardizes public health safety. Previous studies have not shown strong evidence of transmission of EBLs-Ent from milk to humans. However, studies have shown that milk is a source of an outbreak caused by *E. coli* O157:H7 but AMR pattern of strain that caused outbreak was undetermined [162]. Cody et al. [167] reported identical pulsotypes of MDR *Salmonella* Typhimurium in the outbreak, which was later linked to the consumption of raw-milk cheese in California. Recently, Fuenzalida et al. also reported 32 isolates of *Klebsiella* species resistant to 3GCs from a total of 483 isolates collected over 12 years from milk samples in the Wisconsin State [76].

More than 80% of ground beef is obtained from beef cattle in the USA; however, 18% of that is produced from dairy cows sent to slaughterhouses due to their old age or drop in milk yield [168]. Dairy cows constitute about 9.4% of the cattle slaughtered for meat in the USA in 2021 [169]. Due to the extensive use of 3GCs in dairy cattle, ground beef produced from dairy cattle has a high risk of contamination with ESBLs-enteric pathogens [170,171,172]. Thus, undercooked ground beef from culled dairy cows could be a possible source of ESBLs-Ent for humans [146,172]. A study by Iwamoto and his collaborators showed a strong correlation between 3GCs resistant *Salmonella* serotype Newport from humans and ground beef, suggesting possible transmission of 3GCs resistant *Salmonella* to humans [146]. The study showed evidence that dairy cattle are important reservoirs of 3GCs resistant *Enterobacteriaceae* such as *Salmonella* spp., which caused 36 human illnesses in the USA [146].

Similarly, another study linked the 1987 *Salmonella* Newport outbreak to contaminated ground beef from slaughtered dairy cows in California [173]. Again, the source of the recent outbreak of *Salmonella* in Newport that caused 106 illness and several hospitalizations was traced back to ground beef produced from slaughtered dairy cows [171]. The 2019 outbreak of Shiga toxin-producing *E. coli* O103 that affected 209 people in 10 States was linked to ground beef though the AMR pattern of the outbreak strain of *E. coli*, and the source of the ground beef was not reported [174]. Several recent and past multistate outbreaks caused by pathogenic strains of *E. coli* reported by the CDC have been linked to ground beef [174,175,176,177,178,179].

In a nutshell, given the widespread use of ceftiofur in dairy cattle, the increased consumption of raw milk, and undercooked meat, the risk of ESBLs-Ent transmission to humans is high. The resulting infections could be difficult to treat with extended-spectrum cephalosporins or fluoroquinolones. Because both classes of antibiotics are used in dairy cattle and human medicine, co-resistance is also a common phenomenon [8,179,180,181,182]. Though many studies have investigated the types of bacteria present in raw dairy and beef products [153,157,162,175,183,184,185,186] only a few of them looked at the bacterial response to antimicrobial agents [159,187]. Thus, continuous testing of raw milk and beef for AMR bacteria such as ESBLs-Ent and identifying the source of contamination is important to reduce the public health risk that may arise from consumption of these products. In addition, more robust restrictions on selling unpasteurized milk and milk products can help reduce safety hazards arising from their consumption.

### 6.2. Direct Contact (Hand to Mouth) with Dairy Cattle or Their Excretions (Feces, Urine, Milk)

Dairy cattle and dairy products have long been reported as potential reservoirs for 3GCs resistant *Enterobacteriaceae*. Recently, CDC reported a *Salmonella* serotype Heidelberg outbreak affecting 56 people in fifteen States. The outbreak’s source was identified as direct contact with dairy calves. The *Salmonella* isolate involved in the outbreak was MDR, including resistance to 3GCs [188]. Similarly, another study showed strong evidence of transmission of 3GCs resistant *Salmonella enterica* serotype Typhimurium from cattle to humans in the USA [74]. The study employed pulsed field gel-electrophoresis-based analysis of plasmids and beta-lactamases to confirm a link between a domestically acquired ceftriaxone resistant *Salmonella* infection in a child and clonally related isolates from cattle, suggesting cattle could be the source of 3GCs resistant pathogenic and commensal bacteria [74]. Similarly, Gupta et al. [189] also reported 3GCs resistant *Salmonella enterica* serotype Newport isolates with the same pulsotypes and antibiogram patterns from sick dairy cattle, farmworkers, and cattle on the farms in Massachusetts and Vermont, suggesting possible transmission through direct contact.

### 6.3. Fresh Vegetables, Fruits, and Crops

Several fresh vegetables, fruits, and herbs, which are frequently eaten raw, could be contaminated and serve as a source of ESBLs-Ent infection in humans [150,190,191,192,193,194]. This could happen through multiple routes, including when these products are contaminated with manure containing ESBLs-Ent in the field when untreated manure is used as fertilizer, or when these leafy greens are irrigated with water contaminated with feces (Figure 3) [195,196]. Similarly, antimicrobial agents used for crop diseases may contribute to increase pressure on microbes to became resistant or crop could be contaminated from manure used as fertilizer (Figure 3). In the USA, fresh green leafy vegetables have been associated with several outbreaks of enteric pathogens [195,197,198,199,200]. Similarly, fresh vegetables may carry AMR bacteria such as ESBLs-Ent that can enter the human gut when consumed uncooked [201,202,203].

Recently, Liao et al. isolated several ESBLs-*E. coli* strains from ready-to-eat lettuce collected from Northern California [193]. A significant number of the ESBLs-*E.coli* strains isolated from lettuce in this study [193] were previously reported from dairy cattle elsewhere [196], suggesting cow manure’s possible role in contaminating vegetables. The *bla*_CTX-M_ variant was the most frequent ESBL gene detected in *E. coli* isolates obtained from ready-to-eat lettuce. This suggests an increased public health threat linked to vegetable consumption as *bla*_CTX-M_ is increasingly prevalent and expressed by pathogenic strains of *E. coli* in the USA and globally [123,139,193,204]. Similarly, several previous studies also identified ESBLs-Ent such as *E. coli* and *Klebsiella* spp. from USA. green leafy vegetables (e.g., lettuce, spinach, and romaine) ready for human consumption [190,201,203,205]. However, further studies are needed to identify other indirect transmission routes and possible wild animals and birds that may serve as vectors carrying these pathogens among farms and environments as well as sources of contamination of leafy vegetables and fruits.

## 7. Priority Research Gaps That Need to Be Addressed

Based on current literature, we pinpointed the following research gaps that need to be addressed to enhance our understanding and the control of ESBLs-Ent in the USA dairy farms.

A major weakness of current studies is the lack of reliable data on the amount of beta-lactam antibiotics, especially 3GCs used in dairy farms. For example, cephalosporins sales data is often used as an indicator for cephalosporins use which is an unreliable indicator of its use. Furthermore, the sale data does not separately show the amount sold for use in dairy and beef cattle productions. In the absence of these data, it is difficult to assess the impact of their use, develop appropriate interventions, and evaluate the impact of interventions (e.g., the effect of reducing the use of cephalosporins on the prevalence of resistance to cephalosporins). Thus, improving surveillance data on preexisting (baseline) resistance to 3GCs and their use and resistance dynamics after their use is crucial to understanding how antibiotic use may influence antibiotic resistance.Currently, the prevalence of ESBLs-Ent in the USA dairy farms is mostly unknown. For instance, despite the veterinary and public health importance of *Klebsiella* spp., information on its prevalence and the variants of ESBL genes carried by *Klebsiella* spp. isolates from dairy farms are mostly unknown. Further research should address the status of ESBLs-Ent in dairy farms and their potential risk to human health.Among the significant ESBL genes, *bla*_CTX-M_ encoding lineages are establishing themselves as dominant ESBL in *Enterobacteriaceae*, particularly among *E. coli* in the USA dairy farms and across the globe. However, the driver of the successful dissemination of this gene variant is not understood beyond speculation. Understanding the mechanisms for its rapid dissemination in *E. coli* and other members of *Enterobacteriaceae* may help to reduce the emergence and spread of antibiotic resistant commensals and pathogenic strains. Thus, further study is critically important to unravel the mechanisms for widespread dissemination of *bla*_CTX-M_ encoding genes and the bacteria hosting these genesRecently, cases of community-acquired ESBL-Ent infection have been rising in the USA. Despite the widespread speculation, there is a lack of adequate scientific data on the level of ESBLs-Ent transmission from dairy cattle and their farm environments to humans. A further detailed investigation is needed to address the potential transmission of ESBLs-Ent from dairy farms to humans using high-resolution genome sequencing technologies such as WGS in epidemiologically linked settings in a system-based one-health approach. This will help to develop a prudent usage plan and antimicrobial stewardship and infection control policies through one health approach consisting of animal, human and environments.Factors such as antimicrobial usage and farm management practices that may drive the increased prevalence, spread, persistence, and diversity of ESBL-Ent in dairy farms are not adequately investigated in the USA dairy farms. Such studies are needed to enhance our understanding of factors that influence the occurrence and spread of ESBLs-Ent so that evidenced-based control measures can be devised.Archived and contemporary isolates of the members of *Enterobacteriaceae* should be tested to track any temporal changes in the trends (changes) of phenotypic and genotypic resistance to 3GCs over time in the USA dairy farms.

## 8. Conclusions

The widespread use of extended-spectrum cephalosporins in dairy cattle production exposes many healthy cows to antibiotics, resulting in increased selective pressure favoring the propagation of ESBLs-producing bacteria. The growing body of evidence suggests that ESBLs-producing commensal *E. coli*, *Klebsiella,* and *Salmonella* spp. are on the rise in dairy cattle. The rise in ESBLs-organisms in dairy farms can be a significant public health risk, as some of these bacteria are zoonotic and can transmit to humans via various routes. In addition, the resistance genes from commensal bacteria can be transferred horizontally through MGEs to human pathogens, which leads to cross-resistance to antibiotics as the same genetic determinants are responsible for resistance against 3GCs used for the treatment of severe infection in humans. However, the prevalence of ESBLs-Ent in dairy cattle and dairy cattle’s contribution to the burden of ESBLs-Ent infections in humans in the USA is unknown.

Further studies involving temporally and spatially matched samples from dairy cattle, humans and environments in a system-based one-health approach are needed to generate more robust evidence of direct and indirect transmission of ESBLs-producing organisms between humans and dairy cattle using high-resolution techniques such as WGS. Currently, the level of resistance to 3GCs among mastitis-causing pathogens seems low. However, the increase in resistance to 3GCs among commensal *Enterobacteriaceae* can affect dairy cattle’s health by raising the prevalence of MDR bacteria and horizontal exchange of these ARGs. Thus, better knowledge of the major species and their transmission and spread and driving factors of ESBLs-producing organisms and ESBL genes in the USA dairy farms is needed to develop appropriate mitigation strategies.

## Figures and Tables

**Figure 1 antibiotics-11-01313-f001:**
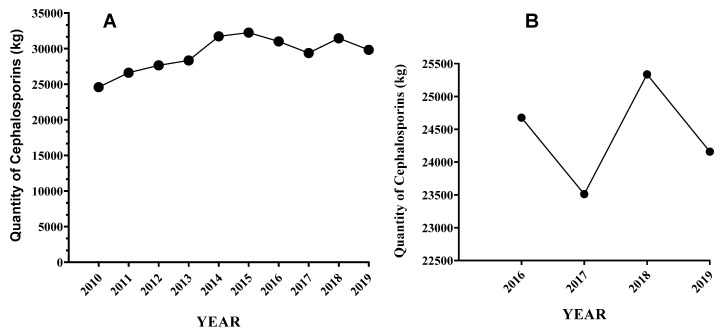
Cephalosporins sold for use in food-producing animals over 10 YEARS (**A**) and for cattle over 4 years (**B**) in the USA [75].

**Figure 2 antibiotics-11-01313-f002:**
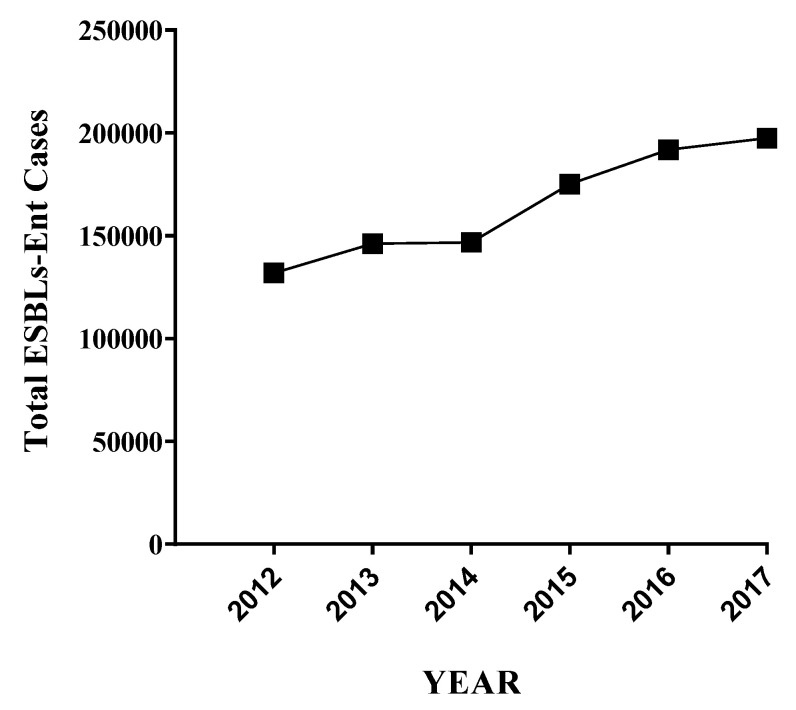
The extended-spectrum beta-lactamases producing *Enterobacteriaceae* infections in humans in the USA [3].

**Figure 3 antibiotics-11-01313-f003:**
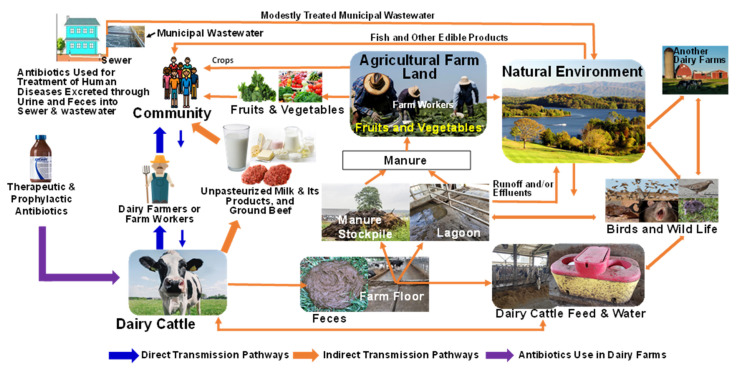
Extended-spectrum beta-lactamases producing *Enterobacteriaceae* and extended-spectrum beta-lactamases genes transmission, spread, and maintenance among dairy farms, environments, and humans within one health settings. Arrows in the figure indicate potential transmission routes, and the thickness of the arrows shows the more likely transmission routes from dairy cattle to humans or vice versa through direct or indirect routes.

**Table 1 antibiotics-11-01313-t001:** Prevalence of extended-spectrum beta-lactamases producing *Enterobacteriaceae* in the USA dairy farms.

Sample from Conventional Farm	Method	Study Design/Population	Pathogens/Prevalence	State/ Region	Reference
Manure, bulk tank milk, manure fertilized soil	CAM, CSM, PCR of ESBL genes	A cross-sectional study on four dairy farms	Prevalence of CTX^r^ *E. coli* was 20.5%, about 36% of BTM isolates were CTX^r^ Over 83% of CTX^r^ isolates carried ESBL genes	TN	[18]
Feces, swabs (pre-evisceration and carcass)	CSM, PCR of ESBL genes	Prospective study on veal calves from four cohorts (farms)	CTX^r^ *E. coli* were 91%, 34% & 19% in feces, pre-evisceration and final carcass swabs, respectively. ESBL genes were detected in 89% of CTX^r^ *E. coli*	OH	[124]
Feces	CSM, WGS	Matched-pair longitudinal study in CEF-treated and non-treated cows	More than 19 CEF^r^ *E. coli* isolates and multiple ESBL genes found	TX, NM	[8]
Feces	Culture	A longitudinal study on cattle with clinical signs of salmonellosis and asymptomatic ones	The proportion of CEF^r^ and CTR^r^ *Salmonella* were 16.5% and 16%, respectively	CA	[125]
Feces	CSM, PCR of ESBL genes	A cross-sectional study on 747 dairy cattle from 25 conveniently selected dairy farms	More than 9% of *E. coli* isolates were CEF^r^, CTX^r^, and CPD^r^. All the 70 *E. coli* isolates carried the ESBL genes	OH	[126]
Feces	CSM, PCR of ESBL genes	On-farm from healthy dairy cattle and dairy cattle submitted for diagnostic purposes	Prevalence of CEF^r^ *Salmonella* isolates were 35.8% and 1.8% among diagnostic and on-farm isolates, respectively		[127]
Feces, lagoons, and milk filters	CSM, PCR of ESBL genes	A retrospective study on *E. coli* isolates banked from a previous survey of 30 dairy farms	The proportions of *E. coli* with ESBL genes were 53.5%, 57.1%, and 50.0% in feces, lagoon, and milk filters from 28 farms, respectively	WA	[57]
Feces	CSM, PCR of ESBL genes	A longitudinal study on 20 dairy heifer calves monthly for five months	About 93% of heifers harbored CEF^r^ *E. coli.* The proportion of CEF^r^ *E. coli* was 100%. ESBL and cephamycinase genes detected	PA	[111]
Milk	Culture, WGS	A cross-sectional study on milk from cows with mastitis from four farms	The prevalence of CEF^r^ *K. pneumoniae* was 2.8%. ESBL genes detected	NY	[122]
Milk	Culture, AST	A retrospective study on 483 *Klebsiella* isolates from milk submitted for testing mastitis	The prevalence of CEF^r^ *Klebsiella* spp. was 6.6%	WI	[76]
Feces	Culture, AST	A cross-sectional study on healthy and sick dairy cattle under different management systems	About 95% and 93% of *E. coli* isolates were CEF^r^ and CTR^r^, respectively	CA	[128]
Composite manure	Culture, AST, and PCR of ESBL genes	A cross-sectional study on 80 dairy farms	CEF^r^ and CTR^r^ *E. coli* were identified in 31.2% and 36.4% of calves, respectively. Similarly, 6.2% and 5% of cows had CTR^r,^ and CEF^r^ *E. coli isolates*, respectively. *E. coli* carrying *bla*_CTX-M_ was identified in about 5% of the farms	PA	[129]
Feces	Culture and AST	A prospective study on *Salmonella* suspected cases over eight months from 2,565 dairy cattle in 412 farms	The prevalence of CEF^r^ *Salmonella* spp. was 60.4%	NY, PA, VT, MA, CT	[130]
Feces from pen floors	CSM and AST	A cross-sectional study on healthy and sick cows from four large-sized dairy farms	More than 51% of *Salmonella* isolates were CEF^r^, and all were susceptible to CTR	SW	[131]
Feces	CSM and AST	A longitudinal study on 110 dairy herds with five times sampling at a two-month interval	Prevalences of CEF^r^ *Salmonella* isolates were 2.4%, 10%, and 10.8% in healthy cows, sick cows, and calves, respectively	NY, MI, MN, WI	[132]

CAM: culture on antibiotic supplemented media; CSM: culture on selective media; CEF^r^: ceftiofur resistant; CTX^r^: cefotaxime resistant; CTR^r^: ceftriaxone resistant; CPD^r^: cefpodoxime resistant; WGS: Whole genome sequencing; AST: antibiotic susceptibility testing; TN: Tennessee; OH: Ohio; TX: Texas; NM: New Mexico; CA: California; WA: Washington, PA: Pennsylvania; NY: New York; WI: Wisconsin; VT: Vermont; MA: Massachusetts; CT: Connecticut; SW: Southwestern Region of USA; MI: Michigan; MN: Minnesota.

**Table 2 antibiotics-11-01313-t002:** Extended-spectrum beta-lactamases encoding genes detected among *Enterobacteriaceae* in the USA dairy cattle.

ESBL Gene Type	Bacteria	State/Region	Sample	Reference
CTX-M-1	*E. coli*	OH, WA, SW	Fecal	[8,29,130]
	*K. pneumoniae*	NY	Mastitic milk	[122]
CTX-M-12	*E. coli*	WA	Fecal	[133]
CTX-14	*E. coli*	OH, WA	Fecal	[24,126]
CTX-15	*E. coli*	OH, WA, SW	Fecal	[8,24,126]
CTX-M-24	*E. coli*	WA	Fecal	[24]
CTX-M-27	*E. coli*	WA, SW	Fecal	[8,24]
CTX-M-32	*E. coli*	SW	Fecal	[8]
CTX-M-55	*E. coli*	WA, SW	Fecal	[8,24]
CTX-M-65	*E. coli*	WA, SW	Fecal	[8,24]
CTX-M-79	*E. coli*	OH	Fecal	[27]
CTX-M	*Salmonella* spp.	Not available	Feces from clinical case	[127]
	*E. coli*	WA	Fecal	[57]
	*E. coli*	TN	Fecal & BTM	[18]
	*E. coli*	OH	Fecal & carcass swabs	[124]
	*E. coli*	PA	Fecal	[129]
SHV	*Salmonella* spp.	Not available	Feces from clinical case	[127]
	*E. coli*	WA	Fecal	[57]
	*K. pneumoniae*	NY	Mastitic milk	[122]
TEM	*E. coli*	OH, WA	Fecal	[126,133]
	*E. coli*	PA	Fecal	[111]
	*Salmonella* spp.	Not available	Fecal	[127]
OXA-27	*E. coli*	WA	Fecal	[133]

TN: Tennessee; WA: Washington; NY: New York; OH: Ohio; PA: Pennsylvania; SW: Southwestern Region of USA.

**Table 3 antibiotics-11-01313-t003:** Beta-lactam antibiotics resistance genes of dairy origin reported by NARMS from 2014 to July 2022.

Beta-Lactam Antibiotic Resistance Gene	Host Bacterium	State	The Proportion of Total Beta-Lactam ARGs of Dairy Source	Time (year)
*bla* _CTXM-27_	*E. coli*	WA, TX, OH and SD	7.2% (7/97)	2017, 2019 and 2021
*bla* _CTXM-55_	*E. coli*	TX, ND	2.1% (2/97)	2019 and 2020
*bla* _CXM-14_	*E. coli*	PA	1% (1/97)	2019
*bla* _CTXM-15_	*E. coli*	TX	1% (1/97)	2018
*bla* _CTXM-65_	*E. coli*	FL	1% (1/97)	2020
*bla* _TEM-1_	*E. coli*	TX, UT, WI, TN, WA, NY, OH, KS, MI, SD, CA, AZ, ID, PA and NE	69.1% (67/97)	2014–2020
*bla* _CMY-2_	*E. coli*	SD, MD, CA, PA, MI, ID, WA and WI	14.4% (14/97)	2018
*bla* _AmpC_	*E. coli*	WI	3.2% (1/31)	2018
*bla* _OXA-2_	*E. coli*	MI	1/97	2019
*bla* _CARB-2_	*E. coli*	CA	1/97	2018
*bla* _SHV-12_	*Salmonella*	CA	1/95	2018
*bla* _CMY-2_	*Salmonella*	UT, WA, WI, CA, TX, ID, UT, CO, AZ, TN and SC	72.6% (69/95)	2015–2020
*bla* _TEM-1_	*Salmonella*	WI, ID, CA, SD, WA, TX, UT, RI, IA and GA	22.11% (21/95)	2015–2019
*bla* _CARB-2_	*Salmonella*	WA, CA	4.21% (4/95)	2014 and 2016

TX: Texas; UT: Utah; WI: Wisconsin; TN: Tennessee; WA: Washington; NY: New York; OH: Ohio; KS: Kansas; MI: Michigan; CA: California; AZ: Arizona; ID: Idaho; CO: Colorado; IA: Iowa; GA: Georgia; RI: Rhode Island; SC: South Carolina; SD: South Dakota; MD: Maryland; MI: Michigan; PA: Pennsylvania; and NE: Nebraska.

## Data Availability

Not applicable.

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
