# Peer review of "Extended-Spectrum Beta-Lactamases Producing Enterobacteriaceae in the USA Dairy Cattle Farms and Implications for Public Health"

_antibiotics, 2022, doi:10.3390/antibiotics11101313_

Round 1
Reviewer 1 Report
The authors provide us with a manuscript reviewing the current state of ESBL-producing Enterobacteriaceae in the United States of America namely in cattle farms. They further discuss the implication of these ESBL-producing bacteria in human public health.
The review is of importance to the field of veterinary and public health in the USA. However, some changes need to be performed for the manuscript to be ready for publication; these comments/suggestions are stated ahead.
General comments:
1. Throughout the manuscript, please replace "United States" and "U.S." with "United States of America" and "U.S.A."
2. The authors should confirm throughout the manuscript the use of abbreviations; sometimes, these abbreviations are stated and then not used anymore by the authors; or the authors sometimes use them and sometimes use the extended term, in an alternation manner. Please rectify this.
3. The authors are very repetitive; please reread the manuscript and consider reducing repetition to facilitate reading and reduce manuscript length. Consider merging information and relocating information to avoid repetition.
4. The authors do not take particular attention to genotyping techniques of extreme importance in the real understanding of bacterial transmission between matrices and hosts; namely, no significant mention of whole-genome sequencing is made with no clear reference to studies performed on this subject; please explore genotypic techniques and their importance on the epidemiology of AMR transmission.
5. The review is indeed focused on Enterobacteriaceae, but only on E. coli and Klebsiella spp.; some mention of Salmonella spp. is also made, but some other important Enterobacteriaceae such as Enterobacter spp. (an ESKAPE pathogen) are never mentioned.
Specific comments:
Line 45 - the authors never clearly stated in previous sentences the meaning of CIA.
Line 46 - the authors already use the abbreviation ARB in line 34; please remove "antibiotic resistance bacteria from this sentence
Line 56 - the authors already use the term third-generation cephalosporin in the manuscript several times but only now the abbreviation is used; please rectify
Line 56 - Ceftiofur is "one of the top 3 most frequently used antibiotics"; what are the other two? what is the percentage of used/quantity used?
Line 60 - The authors already use the term extended-spectrum beta-lactamases in the manuscript several times using the abbreviation; please rectify
Line 64 - The authors already used this abbreviation previously in the manuscript; please rectify
Line 64/65 - this sentence is repetitive to a concept already stated; please remove
Line 70/73 - this sentence is repetitive to one previously stated (line 47/49); please remove one of those
Line 73 - replace "objectives" with "goals" or "aims"
Line 74 - ESBL-ent is an abbreviation previously used; please use it throughout the manuscript
Line 73/76 - the review is focused on the USA scenario; please add this information to the main goals
Figure 1 - Please replace all abbreviations used in figures with their corresponding meaning.
Figure 1 - Consider using the term "One Health" to describe the figure.
Figure 1 - The arrows used to indicate direct and indirect transmission are misleading in the reviewer's opinion; the contact of agricultural workers with contaminated manure is also indirect transmission; the contact with contaminated fruits and vegetables is also indirect transmission; the contact with the contaminated natural environment by fish and other edible products is also indirect transmission; please consider change the figure accordingly.
Line 81 - the mostly used in which context? please clarify
Line 86 - add a reference to the dairy farms' data.
Line 87 - Please use "beta-" instead of "B-"
Line 91 - please state the specific amount (the percentage used or the quantity used).
Line 97/98 - the reviewer does not understand the need to include the Canadian brand name in a review focused on the USA.
Line 99 - the authors do not explore the two intramammary formulations stated here.
Line 109 - the first time a microorganism species is stated, such as S. dysgalactiae, the full name show be stated and not the abbreviation (this means should be stated Streptococcus dysgalactiae. please rectify this throughout the manuscript.
Line 111/115 - some information stated in this paragraph is repetitive to other information previously given; please rectify
Line 121 - please quantity the "frequent use"
Line 123 - please quantify the "primary reason"
Line 124/125 - this information is repetitive with previously stated data.
Figure 2 - please consider starting the graph in 2010 to avoid inducing the readers into mistake.
Line 136 - is the information stated in figure 2 regarding cattle use or food production? please clarify
Figure 3 - please start the graph in 0 kg of cephalosporin to avoid misinterpretation. Maybe make only one graph showing both information (figures 2 and 3) using lines with two colors.
Line 134 and line 137 - information is repetitive.
Line 145 - Please use "Klebsiella spp."
Line 146/147 - quantify "low", "abruptly increase", and "decrease"
Line 147/150 - The authors show plot the percentage of resistance together with the cephalosporin sold to allow the reader a better perception of this variation. Also, the authors refer to figure 3 in this sentence, but the use of cephalosporins in figure 3 is not only related to mastitis treatment; please clarify.
Line 153/155 - this information was already provided previously in the manuscript; please remove it.
Line 155/159 - this information seems repetitive or at least out of order; the relation between antibiotics and diseases was already stated on page 3; please rectify this.
Line 160/167 - states studies related to 3GC resistance; it should come in association with the stated in lines 145/150.
Line 170 - CIA was already used previously; please rectify this.
Line 175 - quantify "reduce".
Line 176 - the authors should consider the relocation of this section to become section 2 since most of the information provided here is not specific to the USA.
Line 213/215 - remove this paragraph since it is repetitive.
Line 216/220 - remove these sentences since they are repetitive.
Table 1 - Table 2 is referred to in line 237, but until here table 1 is not referred to. please rectify this.
Line 240/247 - please remove this paragraph since it is a repetition.
Line 271/278 - remove this paragraph since this information is a repetition of the previously stated information
Line 282 - please explore these "other ARGs"
Line 286 - please state where ("elsewhere") it was found
Line 288 - please explain what are the FIA and FII replicon types
Line 294 - please state which other food-producing animals and which countries have also reported this.
Line 308 - please replace with "Salmonella spp."
Line 309/332 - please consider merging these paragraphs with the paragraph from section 3, namely lines 201/239
Table 1 - it only appears in the main text after table 2; please rectify this. Also, please write the meaning of ESBL instead of using the abbreviation in the title. Please double-check if all abbreviations used are stated in the table's legend. A column with the date of the study should be added.
Lines 338/349 - please consider merging these paragraphs with the paragraph from section 3, namely lines 201/239.
Table 3 - the reviewer does not understand the need to separate table 2 from table 3 since both report ESBL-encoding genes in Enterobacteriaceae in US dairy cattle. Please merge tables.
Lines 360/368 - please consider merging these paragraphs with the paragraph from section 3, namely lines 201/239.
Lines 376/383 - All this information is a repetition of previously stated data; please remove it.
Figure 4 - Please avoid abbreviations in the titles of figures and tables; Is no information available regarding more recent years (from 2018-2021)? Or at least until 2019 as the previous data regarding antibiotic consumption on cattle farms.
Figure 4 - Is no information available on past years? it would be interesting to see a major temporal overview.
Figure 4 - In this figure, the data regarding ESBL-Ent-associated deaths should also be included.
Line 402 - five years not six (2012 does not count since no increase can be clearly stated due to the lack of 2011 data).
Line 416/426 - All this information is a repetition of previously stated data; please remove it.
Line 427/441 - all the information regarding figure 1 should be relocated here (like the information stated in the introduction) and the figure should only appear in this section. This would allow for information aggregation and avoid all the repetition present throughout the manuscript. Please rectify this.
Line 478/480 - this information is repetitive; please remove it.
Line 481/492 - this information should be merged with the previous paragraph.
Line 493/501 - please consider simplifying this paragraph and merging it with lines 523/527 as a conclusion of this subsection.
Lines 543/547 - please remove this information since this review should focus on the USA.
Line 549/559 - please rearrange the paragraph, this information is very repetitive.
Line 569 - please elaborate on which vegetables these contaminations are found.
Line 573 - this should not be point 1 but a paragraph before the point enumeration (this means point 2 should be point 1, etc)
Line 577 - beta
Line 583 - interventions
Line 588/593 - some information exists regarding klebsiella as stated in table 2 (for example) so the authors showed to use the term "mostly unknown" instead of unknown".
Line 595 - Enterobacteriaceae in italicized
Line 618/620 - what do the authors mean by "ESC"? Is no "old" data available regarding ESBL-producing bacteria in USA cattle? The first study was reported more than 2 decades ago; Is the number of studies performed over the past two decades not indicative of an increase in ESBL-producing bacteria in cattle farms? authors should provide this data in a figure as they do for the number of cases in humans.
Author Response
Authors' response to reviewers' comments
Reviewer-1 comments and suggestions
The authors provide us with a manuscript reviewing the current state of ESBL-producing Enterobacteriaceae in the United States of America, namely in cattle farms. They further discuss the implication of these ESBL-producing bacteria in human public health.
The review is of importance to the field of veterinary and public health in the USA. However, some changes need to be performed for the manuscript to be ready for publication; these comments/suggestions are stated ahead
Reviewer-1's general comments (1-5) and Authors' responses:
Reviewer (Rev): Throughout the manuscript, please replace "The United States" and "USA" with "The United States of America" and "USAA.
Author (AU): Accepted and replaced
Rev: The authors should confirm throughout the manuscript the use of abbreviations; sometimes, these abbreviations are stated and then not used anymore by the authors; or the authors sometimes use them and sometimes use the extended term in an alternation manner. Please rectify this.
Au: Agreed and corrected
Rev: The authors are very repetitive; please reread the manuscript and consider reducing repetition to facilitate reading and reduce manuscript length. Consider merging information and relocating information to avoid repetition.
Au: We reviewed the paper and removed or summarized repetitions, and the changes are highlighted in yellow color
Rev: The authors do not take particular attention to genotyping techniques of extreme importance in the real understanding of bacterial transmission between matrices and hosts; namely, no significant mention of whole-genome sequencing is made with no clear reference to studies performed on this subject; please explore genotypic techniques and their importance on the epidemiology of AMR transmission.
Au: We appreciate the reviewer and this is very important comment. But the focus of this manuscript is not reviewing various fingerprinting techniques, including WGS, which are used to study AMR bacterial transmission between animals, humans, and the environment. However, we also mentioned some of the studies that use WGS in table 1). Most importantly, we indicated the importance of using WGS to study ARB and ARGs transmission under the priority research gap section (priority research gap #4).
Rev: The review is indeed focused on Enterobacteriaceae, but only on E. coli and Klebsiella spp.; some mention of Salmonella spp. is also made, but some other important Enterobacteriaceae such as Enterobacter spp. (an ESKAPE pathogen) are never mentioned.
Au: We agree that Enterobacter spp. is an important member of Enterobacteriaceae and might have roles in hosting ESBL-encoding genes. We did not find reports of ESBL-producing Enterobacter spp. from the USA dairy cattle. We would appreciate the reviewer very much if he/she r could recommend published reports of ESBL-Enterobacter spp. in the USA dairy cattle production system that we might have missed.
Reviewer (Rev)1 specific comments and Authors' (Au) response
Rev: Line 45 - the authors never clearly stated in previous sentences the meaning of CIA.
Au: Revised
Rev: Line 46 - the authors already use the abbreviation ARB in line 34; please remove "antibiotic resistance bacteria from this sentence.
Au: Removed
Rev: Line 56 - the authors already use the term third-generation cephalosporins in the manuscript several times but only now is the abbreviation used; please rectify this.
Au: “Third-generation cephalosporins” replaced with "3GCs" throughout the document.
Rev: Line 56 - Ceftiofur is "one of the top 3 most frequently used antibiotics"; what are the other two? what is the percentage of used/quantity used?
Au: The other two most frequently used antibiotics are Cephaprin and Penicillin (for Mastitis) and Penicillin and tetracycline for other diseases of dairy cattle. There is no data on the exact quantity of antibiotics used in dairy cattle. This is one of the important gaps that need to be addressed in the future
Rev: Line 60 - The authors already use the term extended-spectrum beta-lactamases in the manuscript several times using the abbreviation; please rectify this.
Au: Corrected
Rev: Line 64 - The authors already used this abbreviation previously in the manuscript; please rectify
Au: Corrected
Rev: Line 64/65 - this sentence is repetitive to a concept already stated; please remove
Au: Revised
Rev: Line 70/73 - this sentence is repetitive to one previously stated (line 47/49); please remove one of those
Au: Repetition removed and revised
Rev: Replace objectives by aims
Au: "Objectives" replaced by "aims."
Rev: Line 74 - ESBL-ent is an abbreviation previously used; please use it throughout the manuscript
Au: Replaced with ESBL-Ent
Rev: Line 73/76 - the review is focused on the USA scenario; please add this information to the main goals
Au: Added
Rev: Figure 1 - Please replace all abbreviations used in figures with their corresponding meaning.
Au: Replaced
Rev : Figure 1 - Consider using the term "One Health" to describe the figure.
Au: Addressed.
Rev: Figure 1 - The arrows used to indicate direct and indirect transmission are misleading in the reviewer's opinion; the contact of agricultural workers with contaminated manure is also indirect transmission; the contact with contaminated fruits and vegetables is also indirect transmission; the contact with the contaminated natural environment by fish and other edible products is also indirect transmission; please consider change the figure accordingly.
Au: That is correct and we make improved figure 1. Blue arrows show direct transmission pathways, orange arrows show indirect transmission pathways and purple arrow show antibiotic use in dairy farms.
Rev: Line 81 - the mostly used in which context? please clarify
Au: Clarified; in terms of frequency of prescription and use in dairy farms.
Rev: Line 86 - add a reference to the dairy farms' data.
Au: Added
Rev: Line 87 - Please use "beta-" instead of "B-"
Au: Replaced with Beta
Rev: Line 91 - please state the specific amount (the percentage used, or the quantity used).
Au: Specific amount included
Rev: Line 97/98 - the reviewer does not understand the need to include the Canadian brand name in a review focused on the USA.
Au: The Canadian brand name of the antibiotics removed
Rev: Line 99 - the authors do not explore the two intramammary formulations stated here.
Au: The two intramammary formulations of ceftiofur are 1) Ceftiofur hydrochloride suspension for treatment of lactating cows (Brand name: SPECTRAMAST LC) and 2) Ceftiofur hydrochloride suspension for treatment of dry cows (Brand name: SPECTRAMAST DC).
Rev: Line 109 - the first time a microorganism species is stated, such as S. dysgalactiae, the full name show be stated and not the abbreviation (this means should be stated Streptococcus dysgalactiae. please rectify this throughout the manuscript.
Au: All corrected.
Rev: Line 111/115 - some information stated in this paragraph is repetitive to other information previously given; please rectify this.
Au: Only Mastitis is mentioned elsewhere, and other specific diseases for which ceftiofur is used in dairy cattle are not described before.
Rev: Line 121 - please quantity the "frequent use"
Au: The original article did not provide the quantity but was presented by rank (order) as the antibiotic data they collected was skewed. So, the quantity is not available.
Rev: Line 123 - please quantify the "primary reason"
Au: The original articles we cited in the manuscript used a questionnaire survey just to identify qualitatively the main drivers for the use of antibiotics in their farms, for which they responded Mastitis. As a result, they did not have the data on the specific amount of antibiotics used to treat Mastitis. Thus, there were no quantitative data to report.
Rev: Line 124/125 - this information is repetitive with previously stated data.
Au: Revised
Rev: Figure 2 (now figure 1A) - please consider starting the graph in 2010 to avoid inducing the readers into a mistake.
Au: The graph started in 2010
Rev: Line 136 - is the information stated in figure 2 regarding cattle use or food production? please clarify
Au: The information in figure 2 (now figure1A) discusses all food-producing animals, including dairy cattle but figure 3 (now figure 1B) only about cattle.
Rev: Figure 3 - please start the graph in 0 kg of cephalosporin to avoid misinterpretation. Maybe make only one graph showing both information (figures 2 and 3) using lines with two colors.
Au: Figures 2 and 3 are combined into Figures 1A and B.
Rev: Line 134 and line 137 - information is repetitive.
Au: Revised and repetition removed.
Rev: Line 145 - Please use "Klebsiella spp." .
Au: Used
Rev: Line 146/147 - quantify "low", "abruptly increase", and "decrease"
Au: The source article presented the data on a graph and does not show the exact value of prevalence of resistance to ceftiofur for each year. So the exact value is not available for us to present.
Rev: Line 147/150 - The authors show a plot of the percentage of resistance together with the cephalosporin sold to allow the reader a better perception of this variation. Also, the authors refer to figure 3 in this sentence, but the use of cephalosporins in figure 3 is not only related to mastitis treatment; please clarify.
Au: Yes, indeed, cephalosporins are not only used for the prevention and treatment of Mastitis. But Mastitis is the only reason for using first-generation cephalosporins. In addition, the main reason for using third-generation cephalosporins in dairy cattle is also Mastitis (as stated in line 129).
Rev: Line 153/155 - this information was already provided previously in the manuscript; please remove it.
Au: Removed.
Rev: Line 155/159 - this information seems repetitive or at least out of order; the relation between antibiotics and diseases was already stated on page 3; please rectify this.
Au: Revised
Rev: Line 160/167 - states studies related to 3GC resistance; it should come in association with the stated in lines 145/150.
Au: stated, please check references 63 – 67 and references 68 and 21. In this paragraph, we specifically want to show that despite increased usage of ceftiofur for treatment and control of mastitis and other diseases of dairy cattle, the resistance of major mastitis pathogens against ceftiofur is low but resistance of foodborne pathogens associated with dairy farms against ceftiofur is high and also on increasing trend. We are trying to follow the logical order: antibiotic use (ceftiofur), its possible impact on resistance, the status of resistance to ceftiofur among mastitis-causing pathogens, and foodborne pathogens associated with dairy farms and suggested possible alternative strategies.
Rev: Line 170 - CIA was already used previously; please rectify this.
Au: Replaced with CIA
Rev: Line 175 - quantify "reduce."
Au: Rephrased
Rev: Line 176 - the authors should consider the relocation of this section to become section 2 since most of the information provided here is not specific to the USA.
Au: Relocated as section 2.
Rev: Line 213/215 - remove this paragraph since it is repetitive.
Au: The previous similar description of the same concept under Line 206 is removed
Rev: Line 216/220 - remove these sentences since they are repetitive.
Au: The first part (Line 204/215) discusses the classification of the three major ESBLs genes and the general aspects of each gene, including SHV But Line 216/220 gives specific details of SHV, which is not covered under the general description.
Rev: Table 1 - Table 2 is referred to in line 237, but until here, table 1 is not referred to. please rectify this.
Au: Corrected
Rev: Line 240/247 - please remove this paragraph since it is a repetition.
Au: Removed
Rev: Line 271/278 - remove this paragraph since this information is a repetition of the previously stated information.
Au: Revised and repletion removed.
Rev: Line 282 - please explore these "other A.R.G.s"
Au: Addressed
Rev: Line 286 - please state where ("elsewhere") it was found:
Au: Stated
Rev: Line 288 - please explain what are the FIA and FII replicon types.
Au: These are incompatibility (Inc) group F (incF) plasmid replicon alleles
Rev: Line 294 - please state which other food-producing animals and which countries have also reported this.
Au: Addressed
Rev: Line 308 - please replace with "Salmonella spp."
Au: Replaced
Rev: Line 309/332 - please consider merging these paragraphs with the paragraph from section 3, namely lines 201/239.
Au: Section three discusses the mechanism of resistance, but paragraphs from Line 309/332 discuss the status of ESBL-in the USA and are related to Tables 1 and 2. Thus, to address the reviewer's concern, we put it under a separate section (section 5).
Rev: Table 1 - it only appears in the main text after table 2; please rectify this. Also, please write the meaning of ESBL instead of using the abbreviation in the title. Please double-check if all abbreviations used are stated in the table's legend. A column with the date of the study should be added.
Au: The order of mentions to table and 1 are corrected. ESBL is replaced with its expanded version, and all other abbreviations are checked. We don’t have space to add another column, and this table is already a big table. The study data can be found in the cited reference/s if needed.
Rev: Lines 338/349 - please consider merging these paragraphs with the paragraph from section 3, namely lines 201/239.
Au: The first 8 lines (334/345) of this paragraph are related to table 2, and moving this part to the suggested section leaves the table without context. We merged the concept not related to the table (346/350) to section 4( line 307/310), which discusses the spread of ESBL-Ent.).
Rev: Table 3 - the reviewer does not understand the need to separate table 2 from table 3 since both report ESBL-encoding genes in Enterobacteriaceae in US dairy cattle. Please merge tables.
Au: We acknowledge the presence of some overlaps between tables 2 and 3. However, there are two major differences (1) Not all beta-lactamase genes reported in table 3 are ESBL (e.g., blaCMY-2, blaampC, and blaTEM-1); 2) table 3 intends to show the frequency of ESBL genes compared to all other Beta-lactamase encoding genes. This help to lead which genes are predominantly mediating resistance to beta-lactam antibiotics in E. coli and Salmonella spp. in USA dairy farms. Table 2 only focuses on ESBL genes so far reported from the USA dairy farms.
Rev: Lines 360/368 - please consider merging these paragraphs with the paragraph from section 3, namely lines 201/239.
Au: The concept described in this paragraph relates to NARM's report summarized in table 3. Taking this part to other sections will make the table out of context.
Rev : Lines 376/383 - All this information is a repetition of previously stated data; please remove it.
Au: Removed
Rev: Figure 4 - Please avoid abbreviations in the titles of figures and tables; Is no information available regarding more recent years (from 2018-2021)? Or at least until 2019, as the previous data regarding antibiotic consumption on cattle farms.
Au: Yes, we could not find data after the 2017 report and the reviewer may check the recent CDC report ([Source: 2019 AR Threats Report]
Rev: Figure 4 - Is no information available on past years? it would be interesting to see a major temporal overview.
Au: The authors could not find any report of ESBL-Ent infection in the USA before 2012 and after 2017.
Rev: Figure 4 - In this figure, the data regarding ESBL-Ent-associated deaths should also be included.
Au: We have estimated data only for 2017, and thus, we may not be able to draw a graph
Rev: Line 402 - five years, not six (2012 does not count since no increase can be clearly stated due to the lack of 2011 data).
Au: Corrected
Rev: Line 416/426 - All this information is a repetition of previously stated data; please remove it.
Au: Removed
Rev: Line 427/441 - all the information regarding figure 1 should be relocated here (like the information stated in the introduction), and the figure should only appear in this section. This would allow for information aggregation and avoid all the repetition present throughout the manuscript. Please rectify this.
Au: Figure showing routes of spread of ESBL-Ent are relocated, and all description related to it is retained. Repetition is removed from the introduction part.
Rev: Line 478/480 - this information is repetitive; please remove it.
Au: The repeated information is removed.
Rev: Line 481/492 - this information should be merged with the previous paragraph.
Au: Accepted and Merged with the paragraph immediately above it.
Rev: Line 493/501 - please consider simplifying this paragraph and merging it with lines 523/527 as a conclusion of this subsection.
Au: Rearranged and merged
Rev: Lines 543/547 - please remove this information since this review should focus on the USA.
Au: Removed
Rev: Line 549/559 - please rearrange the paragraph, this information is very repetitive.
Au: Rearranged and repetition removed
Rev: Line 569 - please elaborate on which vegetables these contaminations are found.
Au: Elaborated
Rev: Line 573 - this should not be point 1 but a paragraph before the point enumeration (this means point 2 should be point 1, etc).
Au: Accepted and corrected
Rev: Line 577 – beta
Au: Accepted and corrected
Rev: Line 583 – interventions:
Au: Accepted and corrected
Rev: Line 588/593 - some information exists regarding klebsiella, as stated in table 2 (for example) so the authors showed to use the term "mostly unknown" instead of unknown."
Au: Accepted and corrected
Rev: Line 595 - Enterobacteriaceae in italicized.
Au: Accepted and corrected
Rev: Line 618/620 - what do the authors mean by "ESC"? Is no "old" data available regarding ESBL-producing bacteria in USA cattle? The first study was reported more than 2 decades ago; Is the number of studies performed over the past two decades not indicative of an increase in ESBL-producing bacteria in cattle farms? authors should provide this data in a figure as they do for the number of cases in humans.
Au: "ESC" was to mean extended-spectrum cephalosporins. But now we replaced it with "3GCs", which are more specific and relevant to the topic. The authors have summarized reports of ESBL and other beta-lactamase-producing Enterobacteriaceae (Tables 1 and 2). The report the authors summarized in the tables is collected from various studies that use different ESBL detection methods from the clinical and non-clinical samples using different study designs. Thus, with the currently available data, it is not possible to show the actual trend of ESBL-Ent (though it is true that the different reports indicate an increase in ESBL-Ent.)
Reviewer 2 Report
The author provides valuable information that explains the ESBLs studies' current understanding. This review is good to be a guideline for future studies in this field, especially the last summaries of the research gaps will be very helpful. I just got a few points that may be improved.
1. There are some abbreviations that need to give the full name, like CIA in line 45.
2. I found all the figures in the paper lack data from 2021. If you could please include the latest data in the review for better understanding.
Author Response
Reviewer-2 comments and suggestions
The author provides valuable information that explains the ESBLs studies' current understanding. This review is good to be a guideline for future studies in this field, especially the last summaries of the research gaps will be very helpful. I just got a few points that may be improved.
Reviewer-2 specific comments follow (1 and 2):
Reviewer (Rev): There are some abbreviations that need to give the full name, like CIA in line 45.
Authors (Au): Corrected
Rev: I found that all the figures in the paper lack data from 2021. If you could please include the latest data in the review for better understanding.
Au: We agree on the importance of adding the latest data to the figures. However, only data for the years reported on the figures are available.
Reviewer 3 Report
The review entitled “Extended-Spectrum Beta-Lactamases-Producing Enterobacteriaceae in the U.S. Dairy Cattle Farms and Implications for Public Health” by Gelalcha et al. talks about the identification of risk factors for antimicrobial resistance (AMR) in cattle farms of the use and its impact on the public health. I completely agree that reckless use of extended-spectrum cephalosporins in dairy cattle production exposes many healthy cows to the antibiotics resulting in evolutionary pressure on the pathogens for the proliferation of ESBL-producing bacteria. Overall, the study is clear and concise. The introduction is relevant and theory based. Sufficient information about the present study rationale and procedures are provided for the readers. Overall, the results are clear and compelling. The authors make a systematic contribution to the research literature in this area of investigation particularly when the study encompasses different states of America. I would request the authors to revise the review minorly so that it becomes acceptable to the journal
Specific comments follow.
- Please do not add references in the middle of the sentence as in Lines 53,450 and 477. It breaks the flow of reading for the readers
- Line 495. There are as many as 7 references for one sentence. Is it necessary? I’m sure authors can remove some of them which might not be very relevant.
- Make the graphs smaller. They are taking too much of the space.
- I understand that the study is US-based and specific to particular region of the world. Can the authors write a few sentences about the impact of antimicrobial resistance (AMR) on the cattle around the world?
- Line 108 and Line 576 ‘are’ instead of is; ‘study’ instead of ‘studies’
- Reference No. 6. It has been refereed as a recent survey. But it is from 2019. Add the recent survey if possible.
Author Response
Reviewer-3 comments and suggestions
The review entitled "Extended-Spectrum Beta-Lactamases-Producing Enterobacteriaceae in the USA Dairy Cattle Farms and Implications for Public Health" by Gelalcha et al. talks about the identification of risk factors for antimicrobial resistance (AMR) in cattle farms of the use and its impact on the public health. I completely agree that reckless use of extended-spectrum cephalosporins in dairy cattle production exposes many healthy cows to antibiotics resulting in evolutionary pressure on the pathogens for the proliferation of ESBL-producing bacteria. Overall, the study is clear and concise. The introduction is relevant and theory-based. Sufficient information about the present study rationale and procedures are provided for the readers. Overall, the results are clear and compelling. The authors make a systematic contribution to the research literature in this area of investigation, particularly when the study encompasses different states of America. I would request the authors to revise the review minorly so that it becomes acceptable to the journal
Reviewer-3 specific comments (1-5):
Reviewer (Rev): Please do not add references in the middle of the sentence, as in Lines 53,450 and 477. It breaks the flow of reading for the readers.
Authors (Au): The reference is placed at the end of lines 53 and 450. But for 477, we could not put the end at the facts in the sentence are taken from three different sources, and each has to be placed in its respective place.
Rev: Line 495. There are as many as seven references for one sentence. Is it necessary? I'm sure authors can remove some of them which might not be very relevant.
Au: The references are indeed many. However, all those references were cited to show that many studies were conducted on specific aspects of milk-borne pathogens but not only very few on other aspects. The authors feel that this comparison helps readers see the gap and identify where further study is needed.
Rev: Make the graphs smaller. They are taking too much of the space.
Au: Reducing the table size will reduce the information contained in it. We believe the graphs are of the right size and are portrait layouts like the texts.
Rev: I understand that the study is US-based and specific to a particular region of the world. Can the authors write a few sentences about the impact of antimicrobial resistance (AMR on cattle across the globe?
Au: We appreciate this comment. However, as the reviewer indicated, adding the global impact of AMR on our review is out of the scope of this review. In addition, an excellent review was done this year on the global perspective of ESBL on cattle production (Palmeira and Ferreira, 2022; Doi: 10.1016/j.heliyon.2020.e03206).
Rev: Line 108 and Line 576 'are' instead of 'study' instead of 'studies.
Au: Accepted and corrected
Rev: Reference No. 6. It has been referred to as a recent survey. But it is from 2019. Add the recent
Au: Several older surveys (starting from 2007) quantified antibiotic use in USA dairy farms. The use of the term "recent" is relative. We found and added a 2020 survey and modified it as "a "relatively recent survey."
Round 2
Reviewer 1 Report
The reviewer recognizes the general improvement of the manuscript. Authors attend to the majority of this reviewer's comments and suggestions. However, this reviewer would like to state some minor comments:
1) Rev: The authors do not take particular attention to genotyping techniques of extreme importance in the real understanding of bacterial transmission between matrices and hosts; namely, no significant mention of whole-genome sequencing is made with no clear reference to studies performed on this subject; please explore genotypic techniques and their importance on the epidemiology of AMR transmission.
Au: We appreciate the reviewer and this is very important comment. But the focus of this manuscript is not reviewing various fingerprinting techniques, including WGS, which are used to study AMR bacterial transmission between animals, humans, and the environment. However, we also mentioned some of the studies that use WGS in table 1). Most importantly, we indicated the importance of using WGS to study ARB and ARGs transmission under the priority research gap section (priority research gap #4).
REV: In this reviewer's opinion, even due the extended revision of fingerprinting techniques is not the main aim of this paper, an important highlight showed has been made to WGS and its application in the current days. This reviewer also recognizes that this addition should be a decision of the manuscript authors. If the authors do not believe the addion would significantly improve the manuscript, they should not include it.
2) Rev: The review is indeed focused on Enterobacteriaceae, but only on E. coli and Klebsiella spp.; some mention of Salmonella spp. is also made, but some other important Enterobacteriaceae such as Enterobacter spp. (an ESKAPE pathogen) are never mentioned.
Au: We agree that Enterobacter spp. is an important member of Enterobacteriaceae and might have roles in hosting ESBL-encoding genes. We did not find reports of ESBL-producing Enterobacter spp. from the USA dairy cattle. We would appreciate the reviewer very much if he/she r could recommend published reports of ESBL-Enterobacter spp. in the USA dairy cattle production system that we might have missed.
Rev: this reviewer also did not identify any report of ESBL-producing Enterobacter spp. from the USA dairy cattle. However, the occurrence of this bacteria in other countries, such as those in South America should be mentioned and its propagation to the USA discussed since important commercial traits between these countries occurred in high numbers.
Author Response
Reviewer 1:
The reviewer recognizes the general improvement of the manuscript. Authors attend to the majority of this reviewer's comments and suggestions. However, this reviewer would like to state some minor comments:
Rev: In this reviewer's opinion, even due the extended revision of fingerprinting techniques is not the main aim of this paper, an important highlight showed has been made to WGS and its application in the current days. This reviewer also recognizes that this addition should be a decision of the manuscript authors. If the authors do not believe the addion would significantly improve the manuscript, they should not include it.
Au: We totally agree with the reviewer on the importance of WGS and we also added a paragraph about the use of WGS as a powerful tool to study the transmission of ESBL-Ent at the animal-human interface (see lines: 435/448;higlited yellow)
Rev: this reviewer also did not identify any report of ESBL-producing Enterobacter spp. from the USA dairy cattle. However, the occurrence of this bacteria in other countries, such as those in South America, should be mentioned, and its propagation to the USA discussed since important commercial traits between these countries occurred in high numbers.
Au: The ESBL-Ent is still poorly described in South American’s cattle. Few available ESBL-Ent reports from Brazil, Peru, and Chile are limited to E. coli, Klebsiella spp., and Salmonella spp. We only found a thesis study that report one ESBL-enterobacter cloacae from Vampire Bat in Peru [1] and we think further detailed studies are required to determine contribution of Enterobacter spp. to ESBL-Ent in the U.S.A as well as the potential danger of spread from South America to the U. S. A.
- Mendoza, M.V., Identification of antibiotic-resistant enterobacteria in the common vampire bat (Desmodus rotundus) and backyard animals in Department of Lima. 2017, Universidad Peruana Cayetano Heredia: Peru.